# Characterization of Complete Mitochondrial Genomes of the Five *Peltigera* and Comparative Analysis with Relative Species

**DOI:** 10.3390/jof9100969

**Published:** 2023-09-26

**Authors:** Gulmira Anwar, Reyim Mamut, Jiaqi Wang

**Affiliations:** College of Life Sciences and Technology, Xinjiang University, Urumchi 830017, China; gulmira0816@163.com (G.A.); wjqhzh12138@163.com (J.W.)

**Keywords:** *Peltigera*, mitochondrial genome, comparative analysis

## Abstract

In the present study, the complete mitochondrial genomes of five *Peltigera* species (*Peltigera elisabethae*, *Peltigera neocanina*, *Peltigera canina*, *Peltigera ponojensis*, *Peltigera neckeri*) were sequenced, assembled and compared with relative species. The five mitogenomes were all composed of circular DNA molecules, and their ranged from 58,132 bp to 69,325 bp. The mitochondrial genomes of the five *Peltigera* species contain 15 protein-coding genes (PCGs), 2 rRNAs, 26–27 tRNAs and an unidentified open reading frame (ORF). The PCG length, AT skew and GC skew varied among the 15 PCGs in the five mitogenomes. Among the 15 PCGs, *cox2* had the least K2P genetic distance, indicating that the gene was highly conserved. The synteny analysis revealed that the coding regions were highly conserved in the *Peltigera* mitochondrial genomes, but gene rearrangement occurred in the intergenic regions. The phylogenetic analysis based on the 14 PCGs showed that the 11 *Peltigera* species formed well-supported topologies, indicating that the protein-coding genes in the mitochondrial genome may be used as a reliable molecular tool in the study of the phylogenetic relationship of *Peltigera*.

## 1. Introduction

Lichens, symbiotic complexes composed of a fungus and one or more green algae/cyanobacteria, are important parts of their ecosystem [1,2]. In this symbiotic relationship, lichenized fungi (mycobiont) provide water and inorganic salts for algae or cyanobacteria (photobiont/phycobiont). At the same time, they obtain organic compounds from photobionts, forming a mutually beneficial symbiotic relationship [3,4]. The unique structure of lichens enables them to survive in extreme environments where higher plants cannot live, such as mountains and deserts, or the environments of Antarctica [5,6,7]. As a typical symbiont, the main characteristic difference between lichens and other organisms is that they can produce secondary metabolites. These offer good value for food, medicine, medical treatment, environmental monitoring and so on [8,9,10,11]. In lichen, lichenized fungi occupy the main position and determine the morphological characteristics of the lichen. Therefore, lichens are named after the fungal partner in the symbiosis [12].

*Peltigera* Willd. (Lecanoramycetes: Peltigerales) is a lichen-forming genus, and is one of the more widely distributed genera [13]. There is a wide range of morphological and chemical variations in *Peltigera* species, leading to some difficulties in species identification. This has resulted in the taxonomic study of this genus becoming increasingly attractive to many lichenologists [13,14,15,16,17,18]. In 2000, Miadlikowska systematically studied the phylogenetic status of *Peltigera* based on chemical, morphological and large-subunit ribosomal DNA (LSU nrDNA) data and divided the *Peltigera* into eight sections [14]. After that, more researchers began to study the relationship between species in different sections, such as the *Peltigera* section [15,16] and the *Polydocolon* section [17]. Thus far, research on *Peltigera* has been limited mainly to traditional taxonomic studies. Despite the increasing number of genomic data that have been published in recent decades [18,19,20], there are still few studies about the *Peltigera* [21,22,23]. Until now, only six mitochondrial complete genome data of the *Peltigera* could be obtained in the NCBI database, greatly limiting our understanding of the genetic evolution characteristics of the genus.

Mitochondria (Mt), the primary source for the aerobic respiration of cells, are semi-autonomous organelles with their own genomes. The protein-coding genes in the mitochondrial genome can provide abundant sites for phylogeny and help us to understand eukaryotic evolution and genetics [24,25]. With the continuous development of molecular technology, some mitochondrial genes are considered to be universal ‘barcodes’ for the rapid identification of eukaryotes and have become an important molecular marker [26,27]. The size, gene arrangement and structure of fungal mitochondrial genomes vary greatly, even among species of the same genus [28,29,30]. Elucidating the characteristics of fungal mitochondrial genomes, including the changes of these characteristics among different species, will contribute to a comprehensive understanding of the phylogenetic and evolutionary relationships of fungi.

In this study, we sequenced, assembled and annotated five *Peltigera* species. Firstly, the characteristics of five *Peltigera* mitogenomes were revealed. Secondly, by comparing and analyzing the five mitogenomes with the published *Peltigera* mitogenomes, the similarities and differences between the mitochondrial genomes were revealed. Finally, a phylogenetic tree based on protein-coding genes was constructed for the first time to reveal the role of protein-coding genes in the mitochondrial genome in determining the phylogenetic relationship of the genus. This study not only enriches the genome data of the *Peltigera* but also provides information for understanding the genetic evolution of the *Peltigera* species.

## 2. Materials and Methods

### 2.1. Sample Collection and DNA Extraction

Five samples in this study were collected from Xinjiang, China. The species information is shown in Appendix A. All five species were identified by morphological characteristics and ITS sequence (*P. elisabethae:* OR468759; *P. neocanina:* OR473628; *P. canina:* OR470683; *P. ponojensis:* OR468758, *P. neckeri:* OR468736). Specimens were deposited in Herbarium of the College of Life Science and Technology at Xinjiang University in Urumchi, China. Genomic DNA was extracted by fungal DNA extraction kit (Sangon Biotech, Shanghai, China), following the manufacturer’s instructions.

### 2.2. Sequencing, Assembly, and Annotation of Mitochondrial Genomes

Whole genomic sequencing (WGS) of *Peltigera* species was conducted by the DNBSEQ sequencing platform (Shenzhen, China). GetOrganelle v7.4.1 [31] and NOVOPlasty v4.2 [32] were used to assemble *Peltigera* mitogenomes. The complete mitogenomes of *Peltigera* were annotated using MFannot [33], MITOS [34] based on the mitochondrial genetic code 4 [35], and Geseq [36]. Graphical maps of the five *Peltigera* mitogenomes were drawn using OGDraw v1.2 [37].

### 2.3. Sequence Analyses of Mitogenomes

Base composition of the 11 *Peltigera* mitogenomes (the accession numbers are shown in Appendix A) was analyzed using the DNASTAR Lasergene v7.1 (https://www.dnastar.com/, accessed on 11 July 2023). Strand asymmetries of the mitogenomes were calculated according to the following formulas: AT skew = [A − T]/[A + T], and GC skew = [G − C]/[G + C] [38]. Fifteen PCGs (*atp6*, *atp8*, *atp9*, *cytb*, *cox1*, *cox2*, *cox3*, *nad1*, *nad2*, *nad3*, *nad4*, *nad4*L, *nad5*, *nad6*, and *rps3*) in the eleven *Peltigera* mitogenomes were used to detect pairwise genetic distances based on the Kimura-2-parameter (K2P) substitution model, using MEGA v6.06 [39]. The nonsynonymous substitution rates (Ka) and synonymous substitution rates (Ks) of the 15 PCGs (without introns) of the mitogenomes were calculated using DnaSP v6.10.01 [40]. The secondary structure of the tRNA genes in the five *Peltigera* species was predicted by tRNAscan-SE v1.3.1 software [41]. Gene synteny analysis of the five mitogenomes was also calculated using Mauve v2.4.0 [42].

### 2.4. Repetitive Element Analysis

BLASTN searches of the five *Peltigera* mitogenomes against themselves were detected at an E-value of <10^−10^. Tandem repeats in the mitogenomes were identified using the Tandem Repeats Finder software [43]. REPuter [44] was used to detect interspersed repeats in the five mitochondrial genomes: hamming distance was 3, maximum computed repeats was 5000, and minimal repeat size was 30. MISA [45] was used to detect simple sequence repeats (SSRs) in the mitogenomes. The conditions were 10 repeats for mononucleotide, 5 repeats for dinucleotide, 4 repeats for trinucleotide, and 3 repeats for tetranucleotide, pentanucleotide, and hexanucleotide.

### 2.5. Phylogenetic Analysis

We used maximum likelihood (ML) and Bayesian inference (BI) to create phylogenies based on the 14 core genes (*atp6*, *atp8*, *atp9*, *cytb*, *cox1*, *cox2*, *cox3*, *nad1*, *nad2*, *nad3*, *nad4*, *nad4*L, *nad5*, *nad6*), except *rps3* gene that was not annotated in *P. dolichospora*. The *Cairneyella variabilis* was used as an outgroup. Phylosuite used to extract gene [46], MAFFT V7.313 [47] was used for gene alignment, and SequenceMatrix was used for gene concatenation [48]. ModelFinder [49] was used to select the best-fit evolutionary model for the combined gene alignment.

The maximum likelihood (ML) analysis was performed based on the Bayesian information criterion (BIC). Under the Edge-unlinked partition model and ML analysis was performed using IQ-tree V1.6.8 [50], and 1000 ultra-fast bootstrap replications were performed. Bayesian analyses were performed with MrBayes v3.2.7a [51]. Two independent runs with four chains (three heated and one cold) each were conducted simultaneously for 2×10^6^ generations. Each run was sampled every 100 generations. We assumed that stationarity had been reached when the estimated sample size (ESS) was greater than 100 and the potential scale reduction factor (PSRF) approached 1.0. The first 25% of the samples were discarded as burn-in, and the remaining trees were used to calculate Bayesian posterior probabilities (BPP) in a 50% majority-rule consensus tree [52]. Finally, the ML and BI phylogenetic trees were viewed and edited by the network-based Figtree (http://treebioedacuk/software/figtree/, accessed on 23 July 2023).

## 3. Results

### 3.1. Mitogenome Features

In the present study, the complete mitogenomes of *P. elisabethea*, *P. neocanina*, *P. canina*, *P. ponojensis* and *P. neckeri* were assembled, annotated and analyzed. The five *Peltigera* mitogenomes were composed of circular DNA molecules, and their size ranged from 58,132 bp to 69,325 bp (Figure 1). The *P. canina* had the largest mitogenome (69,325 bp) and the *P. neckeri* had the smallest mitogenome (58,132 bp). The average GC contents of the five mitogenomes were very close, with an average of 26.7%. A total of 17 to 21 free-stranding protein-coding genes (PCGs) were detected in the 5 *Peltigera* mitogenomes. All mitogenomes contained 15 PCGs, including *atp6*, *atp8*, *atp9*, *cytb*, *cox1*, *cox2*, *cox3*, *nad1*, *nad2*, *nad3*, *nad4*, *nad4*L, *nad5*, *nad6* and *rps3*. Among the 15 PCGs, *cox1*, *cytb*, *nad1*, *cox2*, *nad4*L and *nad5* contained introns, and ORFs with different sizes were detected in intron regions of the *cox*1 gene. In the five *Peltigera* mitochondrial genomes, intergenic regions and intronic regions accounted for more than 65% of the whole mitogenomes. Genetic regions accounted for 20.04–23.45% of the entire mitogenomes, and the rRNA genes and tRNA genes only accounted for 7.15–8.16% and 2.80–3.34%, respectively (Appendix A).The largest intergenic region was located between *nad4* and *nad1* among the five mitogenomes, with a size of 4202 bp–7673 bp. In addition, one overlapping nucleotide was found in each of the five mitochondrial genomes, located between *nad4*L and *nad5* (Appendix A).

### 3.2. RNA Genes

All of the five *Peltigera* mitogenomes contained two rRNA genes: the small-subunit ribosomal RNA (*rns*) and the large-subunit ribosomal RNA (*rnl*). The lengths of *rnl* genes in the five *Peltigera* mitogenomes ranged from 3188 bp to 3417 bp and 1537 bp to 1539 bp in the *rns* genes. The *rnl* gene had 2–4 introns (Appendix A).

The 5 mitogenomes had 26 tRNA genes, with the exception of *P. ponojensis*, which had 27 tRNA genes. All of the tRNA genes were encoded for the 20 standard amino acids. Further, 26 tRNA genes had the same classical cloverleaf structures in four mitogenomes (Figure 2). Two of the tRNA genes code for leucine, arginine, serine and tyrosine in four of the mitogenomes, with the exception of *P. ponojensis*, with one tRNA gene for tyrosine and three for arginine. Three tRNA genes code for methionine in the five mitogenomes. In the mitogenomes of *P. elisabethae*, *P. neocanina*, *P. canian* and *P. neckeri*, one tRNA gene codes phenylalanine, but two tRNA genes code in *P. ponojensis*. The length of the tRNA genes ranged from 71 bp to 86 bp, and the *trnS* gene was the largest. The lengths of *trnS*, *trnL* and *trnY* were >80 bp, and these tRNAs all contained extra arms, indicating that the size variations of tRNAs were mainly due to size variations in extra arms in the *Peltigera* mitogenomes.

### 3.3. Codon Usage Analysis

We compared the start codons of 15 PCGs in the 11 *Peltigera* species that have been published (Appendix A). Among the 15 PCGs, the *atp6*, *atp8*, *atp9*, *cox1*, *cox3*, *nad1*, *nad2*, *nad3*, *nad4*, *nad4*L, *nad5* and *rps3* genes used ATG as start codons. The start codon of *cytb* was CTG in all observed species, and *cox2* and *nad6* used ATT and ATA as the start codon in the *Peltigera* mitogenomes, except for *P. dolichorrhiza* and *P. polydactylon,* which used GTG as the start codon for the *cox2* gene and ATG for *nad6*.

Codon usage analysis indicated that the codon preferences of the five mitochondrial genomes were highly similar (Figure 3). AGA (for arginase; Arg), UUA (for Leucine; Leu) CCU (proline; Pro), GCU (alanine; Ala) and AGU (Glycine; Gly) were the most frequently used codons in the five mitogenomes.

### 3.4. Repetitive Element Analysis

Through BLASTN searches of the 5 *Peltigera* mitogenomes against themselves, we identified 4, 6, 16, 10 and 6 repeat regions in the *P. elisabethae*, *P. neocanina*, *P. canina*, *P. ponojensis* and *P. neckeri* mitogenomes, respectively (Appendix A). These repeats ranged from 46 to 229 bp. The longest repeats were found in *P. canina*, and the shortest repeat regions were found in *P. neckeri*. In the mitochondrial genome of *P. elisabethae*, *P. neocanina*, *P. canina*, *P. ponojensis* and *P. neckeri* we detected 10, 13, 23, 7 and 10 tandem repeats (Appendix A). The length of repeat units ranged from 4 bp to 124 bp, and the copy number ranged from 1.9 to 12.8. Mononucleotide, dinucleotide, trinucleotide, tetranucleotide, pentanucleotide and hexanucleotide repeats were found in the mitochondrial genome of the five *Peltigera* mitogenomes (Appendix A). Among all of the repeat types, mononucleotide (A/T) and dinucleotide (AT) repeats were the most abundant. A total of 58, 54, 58, 50 and 52 SSRs were detected in the *P. elisabetnae*, *P. neocanina*, *P. canina*, *P. ponojensis* and *P. neckeri* mitogenomes. Among the F (forward repeats), R (reverse repeats), C (complementary repeats) and P (palindromic repeats), forward repeats were found most among the five mitogenomes (Appendix A).

The results of the distribution characteristics of mitochondrial genome repeat sequences show that *P. neocanina,* which had the longest mitochondrial genome, had the most repeat sequences. In addition, repetitive sequences are mainly distributed in intronic and intergenic regions in the mitogenomes (Figure 4).

### 3.5. Variation, Genetic Distance, and Evolutionary Rates of PCGs

Among the 15 protein-coding genes, the lengths of *atp6*, *atp8*, *atp9*, *cox3*, *nad1*, *nad3*, *nad4* and *nad4*L genes were the same in the 11 *Peltigera* mitogenomes (Figure 5, Appendix A). The length of the *cytb* gene was between 1305 bp and 1328 bp, except for *P. rufescens* with a length of 1956 bp. The length of the *cox1* gene varied greatly among the 11 species, and the *rps3* gene was not annotated in *P. dolichospora*. Except for the *cox1* gene, the GC content variation in each gene in the 11 mitochondrial genomes was very small. The GC content of the *cox1* gene was 27% in *P. rufescens* and 33.5% to 33.8% in the other 10 mitogenomes. Among the 15 protein-coding genes, *atp9* had the highest GC content, with an average of 36.1%, and *rps3* had the lowest GC content, with an average of 22.5%. AT skew of the *rps3* gene was positive, AT skews of the remaining 14 genes were negative, but the *cox1* gene in the *P. rufescens* was positive. GC skews of 15 PCGs were variable in the 11 *Peltigera* mitogenomes; among these, *atp6*, *atp8*, *cytb* and *nad2* genes were negative, and other genes were positive. GC skews of the *cox3* gene were negative in *P. neckeri* and *P. polydactylon* but positive in the other mitogenomes.

The nonsynonymous substitution rate, synonymous substitution rate and the Kimura-2-parameter distance were also analyzed in the 11 *Peltigera* mitogenomes (Figure 6). The *cox1* gene had the largest genetic distance, which indicates that the *cox1* exhibited the fastest mutation rate among the 15 PCGs, while the *cox2* gene had the lowest genetic distance, indicating that the *cox2* was highly conserved. Among the 15 PCGs, the *cox1* gene had the highest non-synonymous substitution rate (Ka), and the *cox2* gene had the lowest. The synonymous substitution rate (Ks) of *cox1* was the highest, and that of *cox2* was the lowest. The Ka/ks values of the 15 PCGs were less than 1, which indicates that these genes were selected through purification.

### 3.6. Synteny Analysis

A total of 9 locally collinear blocks (A to I) were detected in 11 *Peltigera* mitogenomes based on the analysis in Mauve (Figure 7). Across the 9 locally collinear blocks detected, B, E and I were found in all 11 mitogenomes. Locally collinear block D was only detected in *P. polydoctylon* and *P. dolichorrhiza*, F and G were found in *P. elisabethea* and *P. neckeri*. Locally collinear block C appeared in nine *Peltigera* mitogenomes, except for *P. elisabethae*, and the size of these regions varied among each species. Among the 11 *Peltigera* species that were analyzed, *P. polydactylon* exhibited 8 locally collinear blocks, except F, while *P. elisabethae* and *P. malacea* exhibited 5 locally collinear blocks. Genome collinearity analyses showed that the position of the PCGs, tRNAs and rRNAs was highly conserved in the *Peltigera* mitogenomes.

### 3.7. Phylogenetic Analysis

A phylogenetic tree of 11 *Peltigera* species was constructed based on the 14 core PCGs (*atp6*, *atp8*, *atp9*, *cytb*, *cox1*, *cox2*, *cox3, nad1, nad2, nad3, nad4, nad4L, nad5, nad6, and rps3*), using both the Bayesian inference (BI) and maximum likelihood (ML) methods. The phylogenetic tree indicates that all branches are well supported (Figure 8). According to the phylogenetic study of Miadlikowska et al., the species used to construct phylogeny were distributed in four sections, while five species in the study were in section D and section E. *P. elisabethae* had a close relationship with *P. neckeri*. *P. ponojensis* and *P. rufescens* formed a sister clade.

## 4. Discussion

In this study, we obtained the mitogenome sequences of five *Peltigera* and analyzed them together with previously published mitochondrial genomes. *P. canina* had the largest mitogenome (69,325 bp), and *P. neckeri* had the smallest mitogenome (58,132 bp) among the five *Peltigera* that were analyzed here. In the fungal mitochondrial genome, the intergenic regions and intronic regions are the main factors affecting the size and variation of the mitochondrial genome [53,54]. The largest mitogenome contained 15 introns with a length of 22,755 bp, accounting for 32.8% of the whole mitogenome. In contrast, the smallest mitogenome had 10 introns, a length of 16,112 bp and accounted for 27.7% of the entire mitogenome. The intergenic regions of *P. elisabethae* were the shortest (17,019 bp, 26.59%) among the five mitochondrial genomes, while the intronic regions were the longest (26,199, 40.94%). The intergenic region and intronic region accounted for 67.53%, 67.17%, 70.01%, 67.39% and 65.05% of the whole mitogenomes in *P. elisabethae*, *P. neocanina*, *P. canina*, *P. ponojensis* and *P. neckeri*, respectively. Variations in intergenic regions are the primary factors underlying the mitogenome size in *Peltigera*, following the intronic region (Appendix A).

The 5 mitochondrial genomes obtained in this study contained 14 core protein-coding genes and 1 *rps3* gene. Furthermore, the 15 PCGs in the *Peltigera* mitogenomes were very conserved in gene order and number but were various in length (Appendix A, Figure 5). Among the 15 PCGs, 8 genes (*atp6*, *atp8*, *atp9*, *cox3*, *nad1*, *nad3*, *nad4*, *nad4*L) had the same length, but the length of the *cox1* gene varied greatly in the 11 *Peltigera* species. Additionally, the Ka, Ks and K2P of the *cox1* gene were the highest in the 15 PCGs, demonstrating that the *cox1* gene had the largest variation in the evolution process. The four mitochondrial genomes, except for *P. neckeri*, have a 504 bp coding region that encodes a hypothetical protein between the *nad6* and *rps3* genes. In addition, the intronic region of the *cox1* gene in the *Peltigera* mitogenome had ORFs of varying lengths, but the way in which introns encode regions is poorly understood. The available mitochondrial genome data are very limited, which makes it impossible for us to conduct more in-depth research on the genomic and evolutionary characteristics of the genus. More genomic data are needed to help us further understand the function of ORFs and hypothetical proteins in fungal mitogenomes.

The tRNA sizes in the five mitochondrial genomes ranged from 71 bp to 86 bp. The existence and size of the extra arm in the *trnS*, *trnL* and *trnY* are the primary factors affecting the size of the tRNA gene. Some studies have found that tRNA mutations affect protein synthesis and various diseases [55,56,57]. Among the G-U, C-U and A-C mismatches, the G-U mismatches are rather unstable; however, G-U mismatches have been found in some fungal genome studies [58,59,60,61]. The predicted tRNA secondary structure showed a large number of G-U mismatches in our study. Different from the *P. elisabethae*, *P. neocanina*, *P. canina* and *P. neckeri* mitochondrial genomes, the *P. ponojensis* mitochondrial genome lacks the *trnY2* gene, and the tyrosine (Tyr) is encoded by only one tRNA gene. In addition, phenylalanine (F) is encoded by two tRNA genes, and only one tRNA gene is encoded in other species. In the secondary structure of other genera of lichens and other fungi, phenylalanine (F) is encoded by only one tRNA gene [62,63].

We detected simple repeats, interspersed repeats and tandem repeats in the mitogenomes (Figure 4), and the results show that the repeat sequences in the mitogenomes were distributed in the intergenic regions. Repetitive sequences in fungal mitogenomes correlate with the mitochondrial gene rearrangement [59]. Synteny analysis found that the protein-coding region was highly conserved in the mitochondrial genome of the *Peltigera*, and the main differences appeared in the non-coding regions (Figure 7). Therefore, the repetitive sequences are likely to be one of the reasons for the large differences between species.

The large variations in the morphology and chemistry of the *Peltigera* species, leading to morphology and anatomy, have not been successfully and quantitively explored in this genus [14]. Although many studies have used multi-gene loci to explore the phylogenetic position of the *Peltigera* [14,15,16,17,21], there are still many species to be discovered, and we need something with richer genetic features to perform phylogenetic analysis on the *Peltigera* [15]. Therefore, we used 14 conserved single-copy protein-coding genes to construct a phylogenetic tree to determine the phylogenetic position of the genus [62,63,64]. According to the previous taxonomic studies of the *Peltigera*, *P. polydactylon* was found in the *Polydactylon* section, *P. malecea* in the *Peltidea* section, *P. elisabethae* and *P. neckeri* in the *Horizontales* section and *P. neocanina*, *P. canina*, *P. ponojensis*, *P. rufescens* and *P. membranacea* in the *Peltigera* section. In our results, *P. elisabethae* and *P. neckeri* are well clustered together, and the five species of the *Peltigera* section are also clustered together with high support. Although our phylogenetic analysis is consistent with the previous results, the data analyzed in this study are limited, and it cannot be confirmed that the protein-coding genes are reliable molecular markers for the phylogenetic study of the *Peltigera*. In future studies, more data are needed to confirm the application and reliability of protein-coding genes in the phylogeny of the *Peltigera*.

## 5. Conclusions

In the present study, we sequenced and assembled five *Peltigera* mitogenomes and compared them with six published *Peltigera* species. The genetic region, intergenic region, intronic region and RNA region were calculated, in which the intergenic region was the main factor affecting the mitogenome size in the *Peltigera*. Gene arrangements were detected in the *Peltigera* mitogenomes, but the protein-coding genes, tRNA genes and rRNA genes were highly conservative in position. Multiple repetitive sequences were found in the five *Peltigera* mitogenomes, including tandem repeats, interspersed repeats and SSRs, and these repetitive sequences are distributed in the intergenic regions and intronic regions. The results of the phylogenetic analysis show that the species of different sections were well clustered together, which indicates the reliability of the protein-coding genes of the mitogenome in the study of the phylogenetic position of the *Peltigera*. This study provides the mitochondrial genome data of the five *Peltigera* and enriches the genome database. In addition, the characteristics of the *Peltigera* mitogenome were revealed, and the similarities and differences among the *Peltigera* species were analyzed, providing information for the genetic evolution of the genus in the future.

## Figures and Tables

**Figure 1 jof-09-00969-f001:**
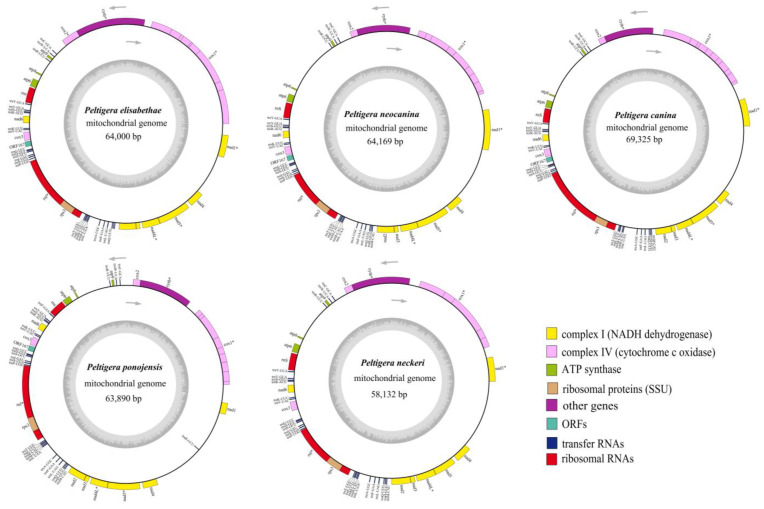
Circular maps of the five *Peltigera* mitogenomes. Genes with different functions are represented by different colors. The genes inside the circle are on the direct strand, and the genes outside the circle are on the reverse strand. Genes with introns are marked with *.

**Figure 2 jof-09-00969-f002:**
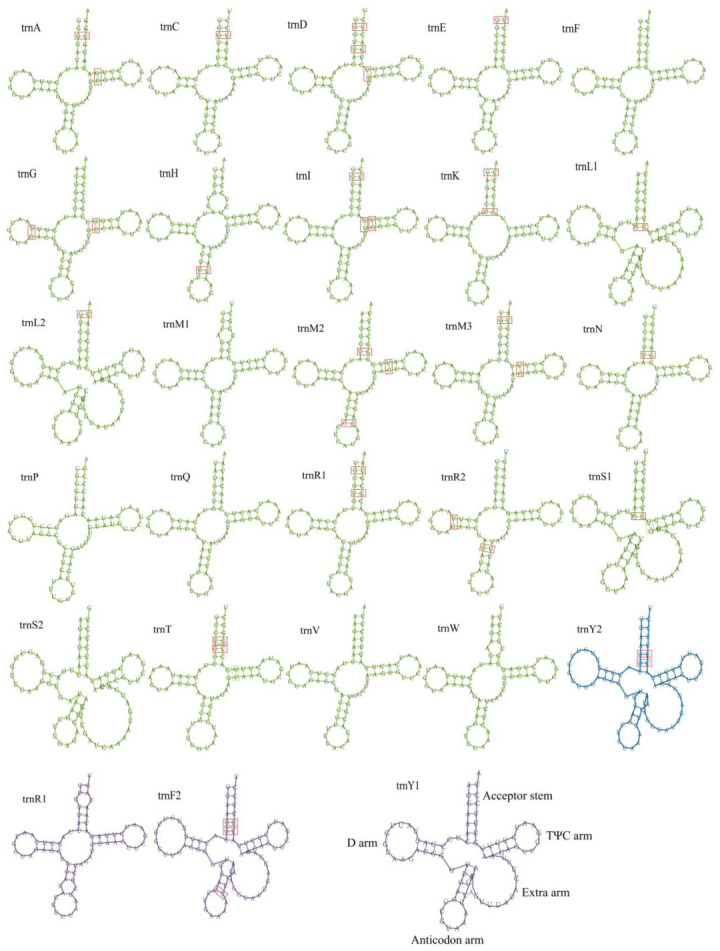
Predicted tRNA secondary structure in the five *Peltigera* mitogenomes. The green tRNAs are present in all five of the mitogenomes, the blue tRNA is present in four *Peltigera* species except *P. ponojensis*, and the purple tRNAs are only present in *P. ponojensis*. The G-U mismatch sites are shown in the red box.

**Figure 3 jof-09-00969-f003:**
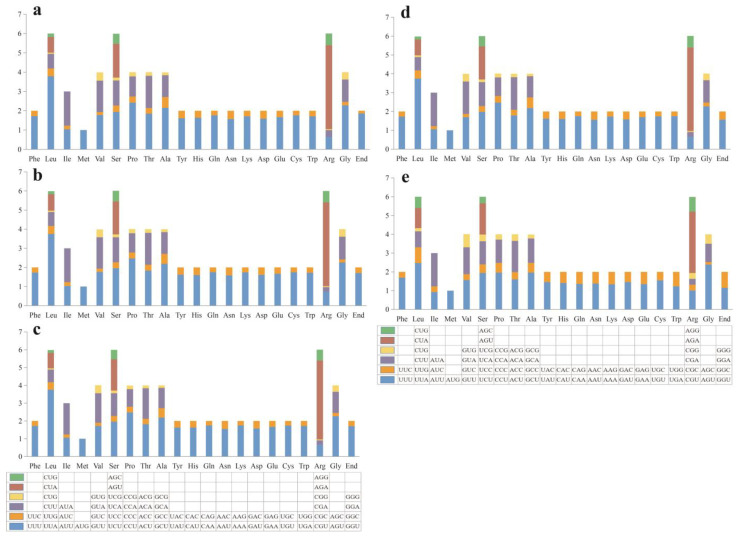
Codon usage analysis of the five *Peltigera* mitogenomes. (**a**) *P. elisabethae*; (**b**) *P. neocanina*; (**c**) *P. canina*; (**d**) *P. ponojensis*; (**e**) *P. neckeri*. The *X*-axis represents the 20 standard amino acids that encode the protein, and below each of the amino acids is the codon that encodes the amino acid. The *Y*-axis represents the frequency of codon usage.

**Figure 4 jof-09-00969-f004:**
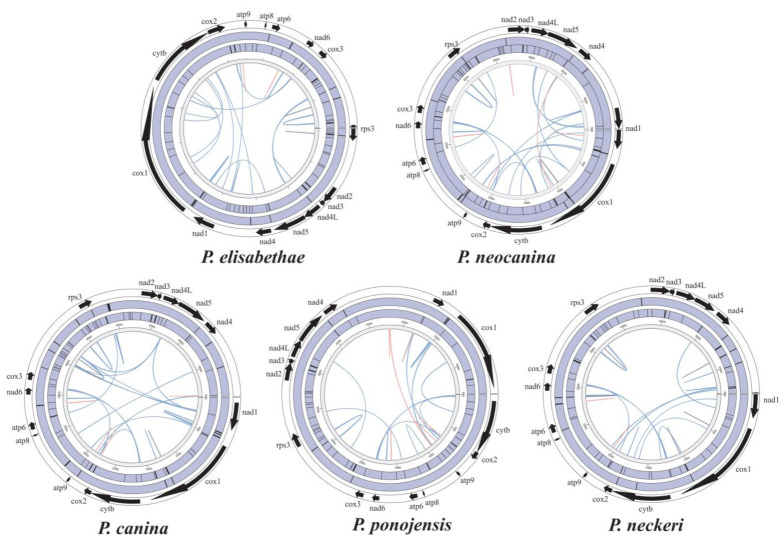
Repeats in the five *Peltigera* mitogenomes. From inside to outside, each circle represents interspersed repeats, simple repeats, tandem repeats and the position of protein-coding genes, respectively. Orange lines indicate forward repeats, blue lines indicate palindromic repeats and grey lines indicate reverse repeats.

**Figure 5 jof-09-00969-f005:**
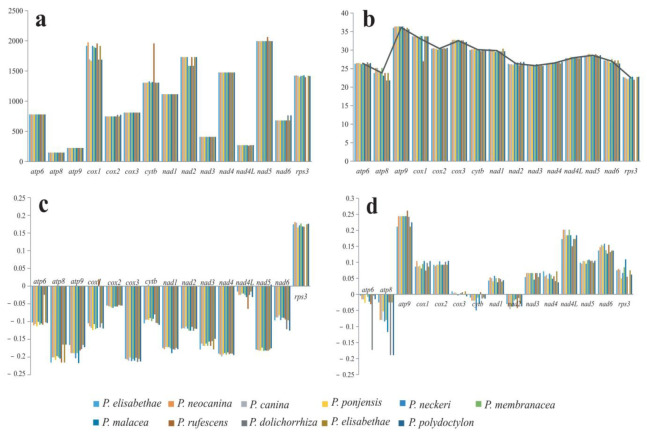
Variation in the length and base composition of each of the 15 protein-coding genes (PCGs) in the 11 *Peltigera* mitogenomes. (**a**) PCG length variation; (**b**) GC content; (**c**) AT skew; (**d**) GC skew.

**Figure 6 jof-09-00969-f006:**
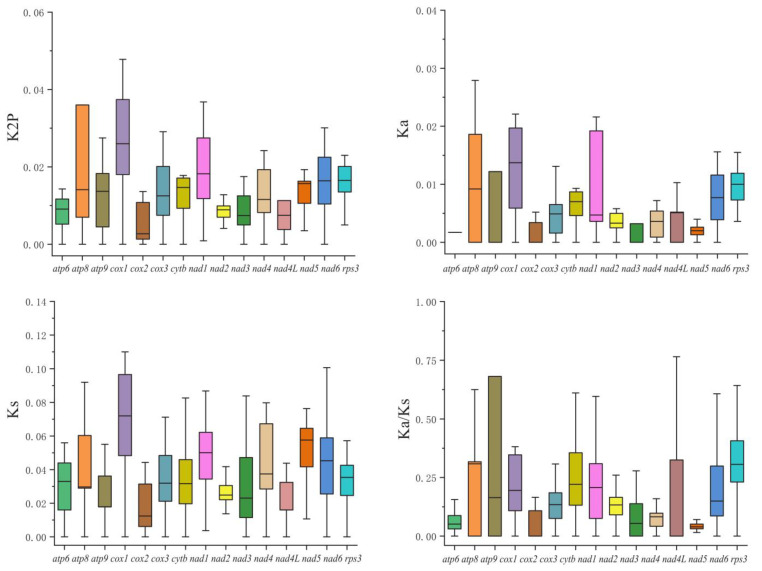
Genetic analysis of 15 protein-coding genes in the 11 *Peltigera* mitogenomes. K2P: the Kimura-2-parameter distance; Ka: non-synonymous substitution rate; Ks: synonymous substitution rate.

**Figure 7 jof-09-00969-f007:**
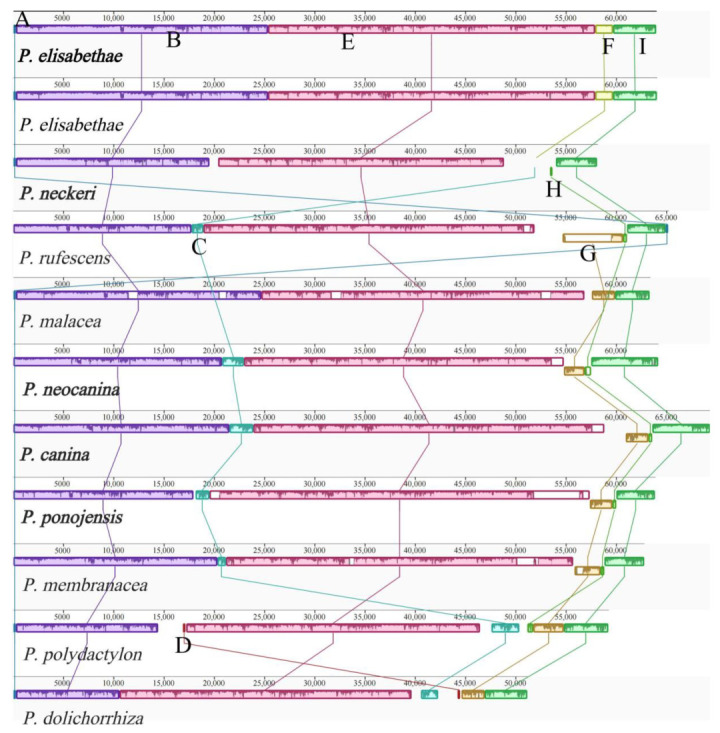
Comparative mitogenomic gene rearrangement analysis of the five *Peltigera* species using Mauve. Locally collinear blocks between different species were represented by the same color blocks. The five species sequenced in this study were displayed in bold.

**Figure 8 jof-09-00969-f008:**
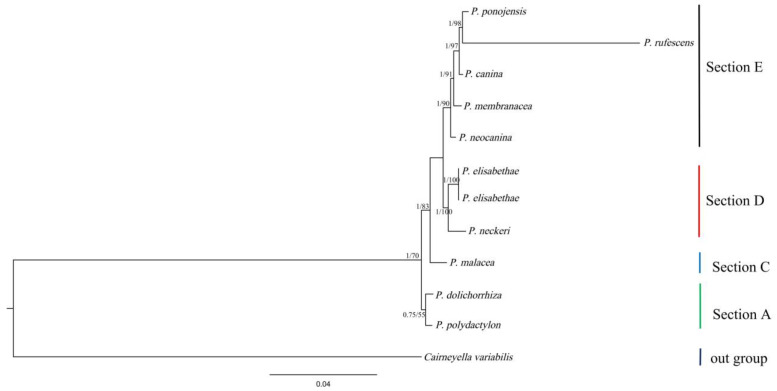
Phylogeny of 11 *Peltigera* species based on the 14 core PCGs using the Bayesian inference (BI) and maximum likelihood (ML) methods, and with *Cyirneyella variabilis* was used as an outgroup. The numbers in the nodes represent bootstrap values (left) and Bayesian posterior probabilities (right). Section labels were lined up with those defined in Miadlikowska. The species and the accession numbers for the mitogenomes used in the phylogenetic analysis are provided in Appendix A.

## Data Availability

The five *Peltigera* mitogenomes of *P. elisabethae*, *P. neocanina*, *P. canina*, *P. ponojensis* and *P. neckeri* were submitted to GenBank under the accession numbers OR343174, OR350404, OR350405, OR350406 and OR350407; and their raw sequencing data were submitted to SRA database under the accession numbers SRR25885166, SRR25885165, SRR25885164, SRR25885163 and SRR25885162, respectively.

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
