# Peer review of "Characterization of Complete Mitochondrial Genomes of the Five Peltigera and Comparative Analysis with Relative Species"

_jof, 2023, doi:10.3390/jof9100969_

Round 1

Reviewer 1 Report

In this study, Anwar et al. describe their study system (Peltigera lichen taxonomy) with relevant background. They describe the problem of limited genomic data within genus, particularly whole genome loci like mitochondrial, despite the improved power for phylogenetic inference. They clearly state what novel science was performed in the study (assembly/annotation of five novel Peltigera mitogenomes and phylogenetic comparison with relatives) and conclude with the impact of the study (provides genomic resources and phylogenetic inference using the mitochondrial PCGs as loci). Their results are thoroughly described, and provide mostly adequate discussion of their results. I have some changes I would like to see, as well as optional suggestions that I think would generally improve the impact/readability of this paper, provided below.

========================

Methods: How were the PCGs extracted for the phylogenetic, K2S and Ka/Ks analyses? I mention below that the unusually high Ka/Ks values for cox1 could be affected by whether the introns, which often fuse with the reading frames of their host gene, are retained when the PCG is pulled out. This would also affect the support values in the phylogeny – if you were to discover that the introns were left in, it could lead to lower inferred support on your branches.

Results: Thorough description of the many analyses performed, including repetitive content, codon-usage bias, tRNA content, and evolutionary analysis of the PCGs.

Discussion:

Adequate disussion of the main findings of the paper, as well as a brief synthesis of the need for studies such as this. However, I would like to see two major points of discussion that are currently missing from the discussion. First is regarding the loss of rps3 in P. dolichospora, if you confirm it to be truly missing. A loss of a PCG, especially one that seems to be ubiquitous in ascomycete mitogenomes, is worthy of some literature review and discussion.

Second, I would like to see a slightly more robust comparison of the phylogenetic relationships revealed by the mitochondrial locus in this study to the Miadlikowska 2001 study. Are the results congruent? What does that say about the reliability of the mitochondrial locus relative to the barcoding loci historically used? Have other whole mitogenome studies found similar patterns?

55: Define ITS

87: genetic code 4. Cite Elzanowski, Andrzej; Jim Ostell (7 July 2010). "The Genetic Codes"National Center for Biotechnology Information. Retrieved 6 May 2013.

114: Mention which of the 15 genes was excluded from the 14 gene alignment and briefly explain why.

143: The PCGs in the introns of cox1, and possibly those of cytb and others, are very likely to be LAGLIDADG homing endonucleases that have been documented in other genera (Aguileta et al., 2014; Kanzi, Wingfield, Steenkamp, Naidoo, & van der Merwe, 2016; Pogoda et al., 2018). Its interesting and validating to see them appear in your results as well, and worth mentioning.

163: This whole paragraph is confusingly-worded. Consider carefully rephrasing.

196: "The copy number ranged from 1.9-12.8" Shouldn't copy number be discrete integers? If so, clarify how this number was arrived at.

215: Please verify using SmartBLAST that cytb in P. rubescens is correct. The much-longer length suggests to me that it has a fused self-splicing group II intron, which has been documented in lichen PCGs (Pogoda et al., 2018; Mukhopadhyay & Hausener 2021).

216: Did you confirm that P. dolichospora is lacking rps3 with ORF-finder and a BLAST using other congeneric references? I would want to see a description of how you ruled out the possibility that GeSeq mistakenly missed this sequence if it remains present in the mitogenome. If it is truly missing in the mitogenome, I would also like to see an attempt to account for the presence of the gene in the nuclear genome.

227: This paragraph describes the genetic distance and the Ka/Ks calculations for each gene. The values for each are fairly believable, however, cox1 seems high and I wouldn't be surprised if the reason is that the coding endonuclease introns whose reading frames are fused with the exons of cox1 were left in for the analysis. This will result in incorrect, high Ka/Ks values. Please ensure that these calculations were performed strictly on the CDS of cox1 with in-frame introns removed. Also add a brief description of that in the methods/results, as well as updating the language in the discussion.

246: I believe the term "locally collinear block (LCB)" is the standard usage for the synteny units output from Mauve, rather than "homologous regions", since one could consider each mitogenome to be entirely homologous with each other. Consider updating this language.

Figures:

Figure 1: This figure would be substantially improved if each genome were oriented to start at the same place. I believe the standard is the 5' end of cox1. It would require very little extra work, since none of your analyses would need to be re-done, just run the new sequence through GeSeq to get new OGDraw figures.

Figure 4: It isn't clear what A-D refer to. Are they the nested circles for each genome? Consider adding labels, or move this figure to SuppMat since it is a bit difficult to interpret.

Figure 5: Label each panel. Also, panel B doesn't make sense as a line graph, since each position on the x-axis isn't connected on a gradient. Change this to a bar graph like the other panels.

Figure 6: Add panel labels so you can just refer to what each graph is showing in the description.

Figure 8: Consider adding Section labels if they line up with those defined in Miadlikowska.

Supplemental Tables:

I am unable to access these materials, since the link provided in the manuscript is currently a placeholder. I would like to review these tables upon resubmission – ensure they are in a high quality so that the manuscript doesn't get sent back for another round of edits on their account. I suggest compiling these tables into one table with multiple tabs and spell check taxon names, etc.

While the science performed in this study is sound, the manuscript requires significant copy edits prior to publication. This includes, but isn't limited to:

1. Singular/plural agreement

2. Fixes to improper capitalization (including section headers, i.e. 3.1. mitogenome 3.1 Mitogenome)

3. Spelling

4. Missing articles (i.e. saying "All five species identified by" instead of "All five species were identified by")

I would like to see substantial edits to the copy for resubmission to improve the readability of this study.

Example copy edits from page 1:

9: Capitalize p. neckeri

14: skew varied of → skew varied among

14: laest → least

17: "14 PCGs" Do you mean 15 PCGs here? If not, clarify why one gene wasn't included

24-25: Lichen, …, is → Lichens,…, are

27: compound → compounds

31: , The → , the

35: after fungi → after the fungal partner in the symbiosis

36: Lecanoramycetess → Lecanoramycetes

44: the Polydocolon section, [17]. → [either finish this list, or end the sentence without the comma]

These are the issues on page 1 -- I have declined to edit the remaining pages for lack of time, but they should be addressed before resubmission.

Author Response

Response to Reviewer 1 Comments

Point 1: Define ITS

Response 1: We uploaded the ITS (Internal Transcribed Spacer) sequences of five species on NCBI and the accession numbers are written in Materials and Methods.

Point 2:genetic code 4. Cite Elzanowski, Andrzej; Jim Ostell (7 July 2010). Retrieved 6 May 2013.

Response 2: This literature has been cited.

Point 3: Mention which of the 15 genes was excluded from the 14 gene alignment and briefly explain why.

Response 3: There are 15 protein-coding genes in the mitogenome of Peltigera, but 14 of them are conserved (Xavier, B. B., Miao, V. P., et al. (2012). Mitochondrial genomes from the lichenized fungi Peltigera membranacea and Peltigera malacea: features and phylogeny. Fungal Biology). Therefore, we selected 14 conserved genes to construct the phylogenetic tree, while the rps3 gene was not included.

Point 4: The PCGs in the introns of cox1, and possibly those of cytb and others, are very likely to be LAGLIDADG homing endonucleases that have been documented in other genera (Aguileta et al., 2014; Kanzi, Wingfield, Steenkamp, Naidoo, & van der Merwe, 2016; Pogoda et al., 2018). Its interesting and validating to see them appear in your results as well, and worth mentioning.

Response 4: As reviewer suggested some of this content is added in the discussion.

Point 5: This whole paragraph is confusingly-worded. Consider carefully rephrasing.

Response 5: This part has grammatical errors that lead to the incoherence of the sentence, and it has been rewritten.

Point 6: "The copy number ranged from 1.9-12.8" Shouldn't copy number be discrete integers? If so, clarify how this number was arrived at.

Response 6:The copy number was obtained by Tandem Repeats Finder, and the specific data are shown in Table 6.

Point 7: Please verify using SmartBLAST that cytb in P. rubescens is correct. The much-longer length suggests to me that it has a fused self-splicing group II intron, which has been documented in lichen PCGs (Pogoda et al., 2018; Mukhopadhyay & Hausener 2021).

Response 7: According to our confirmation, the cytb gene of in P. rufescens was annotated incorrectly, the cox2 gene was annotated in the coding region of the cytb gene.

Point 8: Did you confirm that P. dolichospora is lacking rps3 with ORF-finder and a BLAST using other congeneric references? I would want to see a description of how you ruled out the possibility that GeSeq mistakenly missed this sequence if it remains present in the mitogenome. If it is truly missing in the mitogenome, I would also like to see an attempt to account for the presence of the gene in the nuclear genome.

Response 8: By downloading the sequence of P. dolichospora from NCBI and re-annotating, it was verified that the rps3 gene was not annotated in the P. dolichospora mitogenome, was not missing. We updated the language in the article.

Point 9: This paragraph describes the genetic distance and the Ka/Ks calculations for each gene. The values for each are fairly believable, however, cox1 seems high and I wouldn't be surprised if the reason is that the coding endonuclease introns whose reading frames are fused with the exons of cox1 were left in for the analysis. This will result in incorrect, high Ka/Ks values. Please ensure that these calculations were performed strictly on the CDS of cox1 with in-frame introns removed. Also add a brief description of that in the methods/results, as well as updating the language in the discussion.

Response 9: The genetic distance and the Ka/Ks calculations for each gene were performed strictly on the CDS of cox1 with in-frame introns removed. That was written in the methods/results and discussion.

Point 10: I believe the term "locally collinear block (LCB)" is the standard usage for the synteny units output from Mauve, rather than "homologous regions", since one could consider each mitogenome to be entirely homologous with each other. Consider updating this language.

Response 10: We have made correction according to the reviewer's comments.

Point 11: Figure 1: This figure would be substantially improved if each genome were oriented to start at the same place. I believe the standard is the 5' end of cox1. It would require very little extra work, since none of your analyses would need to be re-done, just run the new sequence through GeSeq to get new OGDraw figures.

Response 11: As the reviewer suggested that each genome have oriented to start at the cox1 gene.

Point 12: Figure 4: It isn't clear what A-D refer to. Are they the nested circles for each genome? Consider adding labels, or move this figure to SuppMat since it is a bit difficult to interpret.

Response 12: From inside to outside, three circles represent three repetitive sequences, and the last circle represents the position of protein-coding genes. We specify in detail what each circle represents in the legend, and A-D were deleted.

Point 13: Figure 5: Label each panel. Also, panel B doesn't make sense as a line graph, since each position on the x-axis isn't connected on a gradient. Change this to a bar graph like the other panels.

Response 13: Considering the reviewer's suggestion, we have added a label to each panel, and in order to show the characteristics of the data more clearly, we changed Figure 2 into a combination of bar graph and line bgraph.

Point 14: Figure 6: Add panel labels so you can just refer to what each graph is showing in the description.

Response 14: The Figure 6 has been changed into the boxplot.

Point 15: Figure 8: Consider adding Section labels if they line up with those defined in Miadlikowska.

Response 15: As the reviewer suggested that Section labels (SectionA, SectionC, SectionD and SectionE) have lined up.

Point 16: Singular/plural agreement

Response 16: We are very sorry for our incorrect writing, these errors have been corrected.

Point 17: Fixes to improper capitalization (including section headers, i.e. 3.1. mitogenome → 3.1 Mitogenome)

Response 17: It has been modified.

Point 18: Spelling

Response 18: We are very sorry for so many small mistakes, at the same time, very grateful to the experts for their valuable advice. Through the editing service of MDPI, we corrected the grammar, spelling and other errors in the article.

Point 19: Capitalize p. neckeri

Response 19: It has been modified.

Point 20: skew varied of → skew varied among

Response 20: It has been modified.

Point 21: laest → least

Response 21: It has been modified.

Point 22: "14 PCGs" Do you mean 15 PCGs here? If not, clarify why one gene wasn't included

Response 22: There are 15 protein-coding genes in the mitogenome of Peltigera, but 14 of them are conserved (Xavier, B. B., Miao, V. P., et al. (2012). Mitochondrial genomes from the lichenized fungi Peltigera membranacea and Peltigera malacea: features and phylogeny. Fungal Biology). Therefore, we selected 14 conserved genes to construct the phylogenetic tree, while the rps3 gene was not included.

Point 23: Lichen, …, is → Lichens,…, are

Response 23: It has been modified.

Point 24: compound → compounds

Response 24: It has been modified.

Point 25: The → , the

Response 25: It has been modified.

Point 26: after fungi → after the fungal partner in the symbiosis

Response 26: We have made correction according to the reviewer's comments.

Point 27: Lecanoramycetess → Lecanoramycetes

Response 27: It has been corrected.

Point 28: the Polydocolon section, [17]. → [either finish this list, or end the sentence without the comma]

Response 28: The comma has been deleted.

I am very sorry for so many small mistakes in the article, and I sincerely thank the review experts for their opinions, and also thank you for taking the time to modify my article and giving me an opportunity to publish my article in the journal of Fungi.

Reviewer 2 Report

The research presented in the MS is backed by a significant financial and infrastructural background. According to the reviewer, the planning and implementation of the research works and the evaluation of the results leaves nothing to be desired .

However, the following minor errors await correction:

Letter and character errors, word repetitions: ad 9., ad 25., ad 31., ad 66., ad 107., ad 133., ad 225., ad 303., ad 461.,

Incomplete citations: ad 40., ad 299. ad 392-393., ad 434.

Scientific Latin names have not been highlighted: ad line 435., ad line 437., ad line 450.

Foreign language insertions are recommended in italics: ad 454, ad 456.

Ad Fig. 4 and fig. 5.: the letter codes of the figures are missing

Incomplete sentence / editing error: ad 324.

Moderate editing of English language maybe useful.

Author Response

Response to Reviewer 2 Comments

Point 1: Letter and character errors, word repetitions: ad 9., ad 25., ad 31., ad 66., ad 107., ad 133., ad 225., ad 303., ad 461.

Response 1: We have made correction according to the comments.

Point 2: Incomplete citations: ad 40., ad 299. ad 392-393., ad 434.

Response 2: We have made correction according to the reviewer's comments.

Point 3: Scientific Latin names have not been highlighted: ad line 435., ad line 437., ad line 450.

Response 3: These mistakes have been corrected.

Point 4: Foreign language insertions are recommended in italics: ad 454, ad 456.

Response 4: I 'm very sorry I couldn't find accurately the need to modify.

Point 5: Ad Fig. 4 and fig. 5.: the letter codes of the figures are missing

Response 5: Fig 4 and Fig 5 have been modified.

Point 6: Incomplete sentence / editing error: ad 324.

Response 6: It has been modified according to the reviewer's comments.

I am very sorry for so many small mistakes in the article, and I sincerely thank the review experts for their opinions, and also thank you for taking the time to modify my article and giving me an opportunity to publish my article in the journal of Fungi.

Reviewer 3 Report

The manuscript "Characterization complete mitochondrial genomes of the five Peltigera and comparitive analysis with relative species" presents a well-performed phylogenetics analysis and five new Peltigera species. I will recommend several points for improving the manuscript, especially in the presentation of the image and the discussion of the mitogenome. phylogeny. 

Major points:

1) Please check the manuscript for typos. There is one in the title, others in the text

Title - typo "comparitive analysis"

Line 107 - typo "fivenmitochondrial"

Line 163 - typo in "structers"

Line 84 - Rephrase better: "We obtained raw data >6 G". Use Gb instead and make more clear this sentence

2) Scientific language

* Line 42 - change "sections" to "clades"

* Lines 107, 316, 247, 363 - Change "scattered repeats" to "interspersed repeats"

3) Data release

* You have sequenced the genomes of 6 new species. Could you please release the raw data through SRA - NCBI?

* Line 84 - Please provide the N50, max, and min length for the sequenced reads. Add that in the Supplementary. The technology is relatively new and the readers are interested to know more about that

4) Figure quality

* Please describe better the Figures in their legend. There are several Figure details that are unclear to the reader. See the breakdown in Figure

Figure 2 - You mention that "26 tRNA genes were the same classical cloverleaf structers in four mitogenomes". The legend says "Predicted tRNA secondary structure in the five Peltigera mitogenomes". What is the Figure actually showing? Are these the conserved tRNAs for four species? What about the tRNAs that are not the same in the other mitogenome?

The Figure has red squares for highlighting and I am not sure what is that showing. Please explain in the legend.

Figure 3 - Please provide more description of the Figure in the Legend. Tell about the Table and the color scheme. It was not very easy to understand it at a first glance

Figure 5 - I don't see where you cite it in the text. The Figure is only described in the discussion. Include its description in section 3.5

Furthermore, please remove the legend from the bottom of every Figure. It's the same for all the panels from the Figure so you can keep only ONE of that. The Figure is redundant right now

Again, add more details in the legend

Figure 6 - Is each panel showing the mean value for the 11 species? Can you please provide the individual values through a different type of plot, such as a boxplot? Please see the image in this paper - https://www.frontiersin.org/files/Articles/529593/fmicb-11-00617-HTML/image_m/fmicb-11-00617-g006.jpg

Figure 7 - Why are you showing only 5 of the genomes? what happened to the others? The image looks like a screenshot, there is an awkward sign "R" on the y-axis which I would get rid of

Figure 8 - Explain what the numbers at the nodes represent. Which one is BI or ML? Provide more details in the legend.

The length of all the branches is not shown you are displaying the tree as a cladogram but you still keep the reference distance on the bottom (0.04). Can you please plot the actual distance that you've obtained? That will make the Figure correct

Discussion

You use the mitogenome for phylogeny which is however only one part of the evolutionary story. People use nuclear genomes now that we have the entire genome sequenced. Could you please mention in the discussion what are the limitations of using only the mitogenomes for phylogeny? You mention that "This indicated that mitochondrial protein-coding genes can be used as molecular markers for phylogenetic analysis of the genus" but you don't have enough species to better resolve the tree. Could you please tone down this statement and also include more discussion about the limitations of your approach? Thank you in advance

Overall English language is okay. Please check for typos

Author Response

Response to Reviewer 3 Comments

Point 1: Please check the manuscript for typos. There is one in the title, others in the text

Title - typo "comparitive analysis"

Line 107 - typo "fivenmitochondrial"

Line 163 - typo in "structers"

Line 84 - Rephrase better: "We obtained raw data >6 G". Use Gb instead and make more clear this sentence

Response 1: We are very sorry for so many small mistakes, at the same time, very grateful to the experts for their valuable advice. Through the editing service of MDPI, we corrected the grammar, spelling and other errors in the article.

Point 2: Change “sections” to “clades”

Response 2: Miadlikowska used “section” in the article entitled “Phylogenetic revision of the genus Peltigera (lichen-forming Ascomycota) based on morphological, chemical, and large subunit nuclear ribosomal DNA data”. According to the definition in Miadlikowska, we used “section”.

Point 3: Lines 107, 316, 247, 363 - Change "scattered repeats" to "interspersed repeats"

Response 3: We have made correction according to the reviewer's comments.

Point 4: You have sequenced the genomes of 6 new species. Could you please release the raw data through SRA - NCBI?

Response 4: In this study, we sequenced the mitochondrial genomes of five species, but only the annotation file was uploaded to NCBI, and the raw data was not uploaded. We have contacted NCBI to request the release of data. If the reviewer asks us to upload the raw data to SRA, we will upload it when the submission service is available (Now, NCBI SRA submission services are unavailable due to scheduled maintenance).

Point 5: Please provide the N50, max, and min length for the sequenced reads. Add that in the Supplementary. The technology is relatively new and the readers are interested to know more about that.

Response 5: The data in this study had been sequenced using the PE150 strategy of Paired-end sequencing. That means the sequencing reads are all 150 bp. But as the reviewer suggested the N50, max, and min length for the sequenced reads have been provided in the Supplementary Table 10.

Point 6: Figure 2 - You mention that "26 tRNA genes were the same classical cloverleaf structers in four mitogenomes". The legend says "Predicted tRNA secondary structure in the five Peltigera mitogenomes". What is the Figure actually showing? Are these the conserved tRNAs for four species? What about the tRNAs that are not the same in the other mitogenome? The Figure has red squares for highlighting and I am not sure what is that showing. Please explain in the legend.

Response 6: We are very sorry for our negligence. Figure 2 shows the tRNA secondary structure in the five Peltigera mitogenomes. tRNA secondary structures of different species are distinguished by different colors and explained in the legend. The red squares represent G-U mismatch sites.

Point 7: Figure 3 - Please provide more description of the Figure in the Legend. Tell about the Table and the color scheme. It was not very easy to understand it at a first glance

Response 7: We have described the Figure in detail according to the reviewer's comments.

Point 8: Figure 5 - I don't see where you cite it in the text. The Figure is only described in the discussion. Include its description in section 3.5

Response 8: We are very sorry for our negligence. The Figure 5 are cited in section 3.5.

Point 9: Figure 6 - Is each panel showing the mean value for the 11 species? Can you please provide the individual values through a different type of plot, such as a boxplot?

Response 9: We have changed Figure 6 into the boxplot according to the reviewer's comments.

Point 10: Why are you showing only 5 of the genomes? what happened to the others? The image looks like a screenshot, there is an awkward sign "R" on the y-axis which I would get rid of

Response 10: We added the remaining six species and performed a collinearity analysis on 11 species. The picture is exported from Mauve, not a screenshot."R" has been deleted.

Point 11: Figure 8 - Explain what the numbers at the nodes represent. Which one is BI or ML? Provide more details in the legend.

Response 11: More discription were written in the legend according to the reviewer's comments.

Point 12: You use the mitogenome for phylogeny which is however only one part of the evolutionary story. People use nuclear genomes now that we have the entire genome sequenced. Could you please mention in the discussion what are the limitations of using only the mitogenomes for phylogeny? You mention that "This indicated that mitochondrial protein-coding genes can be used as molecular markers for phylogenetic analysis of the genus" but you don't have enough species to better resolve the tree. Could you please tone down this statement and also include more discussion about the limitations of your approach? Thank you in advance

Response 12: Although our phylogenetic results are consistent with the results of previous studies, this conclusion ("This indicated that mitochondrial protein-coding genes can be used as molecular markers for phylogenetic analysis of the genus") cannot be drawn only by relying on the lack of these data. We have modified these contents in the discussion.

As reviewer suggested some of this content is added in the discussion.

I am very sorry for so many small mistakes in the article, and I sincerely thank the review experts for their opinions, and also thank you for taking the time to modify my article and giving me an opportunity to publish my article in the journal of Fungi.

Round 2

Reviewer 3 Report

Thank you for all the changes and improvements on the manuscript. Please see the following suggestions (in bold)

Point 4: You have sequenced the genomes of 6 new species. Could you please release the raw data through SRA-NCBI?

Response 4: In this study, we sequenced the mitochondrial genomes of five species, but only the annotation file was uploaded to NCBI, and the raw data was not uploaded. We have contacted NCBI to request the release of data. If the reviewer asks us to upload the raw data to SRA, we will upload it when the submission service is available (Now, NCBI SRA submission services are unavailable due to scheduled maintenance).

Point 5: Please provide the N50, max, and min length for the sequenced reads. Add that in the Supplementary. The technology is relatively new, and the readers are interested to know more about that.

Response 5: The data in this study had been sequenced using the PE150 strategy of Paired-end sequencing. That means the sequencing reads are all 150 bp. But as the reviewer suggested the N50, max, and min length for the sequenced reads have been provided in the Supplementary Table 10.

1) Could you please release the raw data on NCBI? It's a good practice to share all the results, given the nature of your academic research. Please carry on with the submission and share the proof that you have submitted the raw data to SRA

2) Could you please make clear what type of data you have used in the study? On line 82, could you add the info about the PE read length? I was not quite sure what technology you have applied there

Thank you in advance for adding all these improvements. It looks neater now.

Author Response

Response to Reviewer 3 Comments

Thank you for your letter and the reviewer’s comments.

Point 1: Could you please release the raw data on NCBI? It's a good practice to share all the results, given the nature of your academic research. Please carry on with the submission and share the proof that you have submitted the raw data to SRA

Response 1: The raw data has been uploaded to the SRA, P. elisabethae: SRR25885166; P. neocanina: SRR25885165; P. canina: SRR25885164; P. ponojensis: SRR25885163, P. neckeri: SRR25885162.

Point 2: Could you please make clear what type of data you have used in the study? On line 82, could you add the info about the PE read length? I was not quite sure what technology you have applied there

Response 1: Sequencing library was generated using NEBNext® UltraTM DNA Library Prep Kit for Illumina (NEB, USA, Catalog #: E7370L) following manufacturer’s recommendations and index codes were added to each sample. Briefly, genomic DNA sample was fragmented by sonication to a size of 350 bp. Then DNA fragments were endpolished, A-tailed, and ligated with the full-length adapter for Illumina sequencing, followed by further PCR amplification. After PCR products were purified by AMPure XP system (Beverly, USA). Subsequently, library quality was assessed on the Agilent 5400 system(Agilent, USA) and quantified by QPCR (1.5 nM). After library quality control, different libraries were pooled based on the effective concentration and targeted data amount. 5'-end of each library was phosphorylated and cyclized. Subsequently, loop amplification was performed to generate DNA nanoballs. These DNA nanoballs were finally loaded into flowcell with DNBSEQ-T7 for sequencing in Shenzhen Huitong biotechnology Co. Ltd (Shenzhen, China).

Paired-end reads substantially facilitate assemblies of all genome sizes. Paired-end reads can also help resolve differences among repeat regions and thus can be used in transcriptome projects to distinguish family members as well as identify alternative splicing. The most commonly run paired-end reads in the Core are PE100 and PE150 (HiSeq), and  PE250 (Miseq and Rapid Mode Hiseq), and PE300 (MiSeq).